# Immune-Based Therapy in Triple-Negative Breast Cancer: From Molecular Biology to Clinical Practice

**DOI:** 10.3390/cancers14092102

**Published:** 2022-04-23

**Authors:** Francesca Carlino, Anna Diana, Antonio Piccolo, Anna Ventriglia, Vincenzo Bruno, Irene De Santo, Ortensio Letizia, Ferdinando De Vita, Bruno Daniele, Fortunato Ciardiello, Michele Orditura

**Affiliations:** 1Department of Precision Medicine, Division of Medical Oncology, University of Campania Luigi Vanvitelli, 80131 Naples, Italy; antonio.piccolo2@studenti.unicampania.it (A.P.); anna.ventriglia@unicampania.it (A.V.); vincenzo.bruno@unicampania.it (V.B.); ferdinando.devita@unicampania.it (F.D.V.); fortunato.ciardiello@unicampania.it (F.C.); michele.orditura@unicampania.it (M.O.); 2Medical Oncology Unit, Ospedale Ave Gratia Plena, San Felice a Cancello, 81027 Caserta, Italy; irene.desanto@ospedalesancarlo.it (I.D.S.); ortensio.letizia@aslcaserta.it (O.L.); 3Medical Oncology Unit, Ospedale del Mare, 80147 Naples, Italy; anna.diana@unicampania.it (A.D.); b.daniele@libero.it (B.D.)

**Keywords:** triple-negative breast cancer, immunotherapy, biomarkers, immune checkpoints inhibitors, treatment, resistance

## Abstract

**Simple Summary:**

Triple-negative breast cancer has been historically considered an orphan disease in terms of therapeutic options. To date, chemotherapy is still the mainstay of treatment both in the early and metastatic settings. Recent advances in the genomic and immunologic fields have revealed the molecular complexity and the immune profile of this breast cancer subtype, resulting in the development of novel therapeutic strategies, including immunotherapy. This review provides a comprehensive overview of the immune system and the different immunotherapeutic drugs approved or under investigation for the treatment of triple-negative breast cancer, with a focus on the potential strategies to enhance immune responses and overcome mechanisms of resistance.

**Abstract:**

Triple-negative breast cancer (TNBC) has been considered for many years an orphan disease in terms of therapeutic options, with conventional chemotherapy (CT) still representing the mainstay of treatment in the majority of patients. Although breast cancer (BC) has been historically considered a “cold tumor”, exciting progress in the genomic field leading to the characterization of the molecular portrait and the immune profile of TNBC has opened the door to novel therapeutic strategies, including Immune Checkpoint Inhibitors (ICIs), Poly ADP-Ribose Polymerase (PARP) inhibitors and Antibody Drug Conjugates (ADCs). In particular, compared to standard CT, the immune-based approach has been demonstrated to improve progression-free survival (PFS) and overall survival (OS) in metastatic PD-L1-positive TNBC and the pathological complete response rate in the early setting, regardless of PD-L1 expression. To date, PD-L1 has been widely used as a predictor of the response to ICIs; however, many patients do not benefit from the addition of immunotherapy. Therefore, PD-L1 is not a reliable predictive biomarker of the response, and its accuracy remains controversial due to the lack of a consensus about the assay, the antibody, and the scoring system to adopt, as well as the spatial and temporal heterogeneity of the PD-L1 status. In the precision medicine era, there is an urgent need to identify more sensitive biomarkers in the BC immune oncology field other than just PD-L1 expression. Through the characterization of the tumor microenvironment (TME), the analysis of peripheral blood and the evaluation of immune gene signatures, novel potential biomarkers have been explored, such as the Tumor Mutational Burden (TMB), Microsatellite Instability/Mismatch Repair Deficiency (MSI/dMMR) status, genomic and epigenomic alterations and tumor-infiltrating lymphocytes (TILs). This review aims to summarize the recent knowledge on BC immunograms and on the biomarkers proposed to support ICI-based therapy in TNBC, as well as to provide an overview of the potential strategies to enhance the immune response in order to overcome the mechanisms of resistance.

## 1. Introduction

### Rationale of Immune-Based Therapy in Breast Cancer

The complex interaction between cancer and immune cells and the understanding of the immune escape mechanisms led to the development of the immuno-oncology field.

In physiological conditions, there is a balance between proinflammatory and anti-inflammatory signaling regulated by immune checkpoints to prevent autoimmunity. These immune checkpoints are a set of inhibitory and stimulatory pathways that directly affect the function of immune cells. During cancer progression, the occurrence of several genomic mutations leads to the production of tumor neoantigens, which, in turn, could be recognized and destroyed by the immune system as being perceived as non-self. This dynamic process, called cancer immunoediting, is composed of three sequential phases (elimination, equilibrium, and escape), whereby the host immune cells activate the innate and adaptive responses to protect against tumor formation and shape tumor immunogenicity.

However, one of the hallmarks of cancer is the ability of malignant clones to evade immune-mediated destruction by multiple mechanisms, including impaired antigen presentation, upregulation of negative regulatory pathways and the recruitment of immunosuppressive cells populations. In particular, regulatory T (Tregs) cells in the tumor microenvironment (TME) display strong immune suppressive activity and are thus able to inhibit antitumor immune responses by means of cytokines activating inhibitory immune checkpoints.

In the last few years, three types of immunotherapeutic strategies have been employed and classified into: “passive”, including the infusion of monoclonal antibodies (moAbs), i.e., IgG isotypes that bind and neutralize a target tumor associated with or specific antigen yielding the lysis of cancer cells, or the systemic administration of recombinant cytokines; “active”, consisting of the administration of ICIs and vaccines, and “adoptive”, which exploits immune system cells to eliminate cancer cells, such as autologous T cell-based therapy [1] (Figure 1).

The outcome of ICI-based therapy largely depends on the immunogenic nature of the tumor, as demonstrated by the remarkable response and survival gain obtained in melanoma and small cell lung cancer treatment [2,3].

The introduction of immunotherapy in the treatment of breast cancer (BC) has been markedly delayed due to the low mutation rate and weak immunogenic potential compared to other malignancies.

However, recent evidence has demonstrated that the expression of immune genes, cytokines and the composition of the immune infiltrates are involved in BC occurrence and progression, supporting research efforts in the immunotherapy field [4,5].

A BC immune landscape is characterized by different degrees of immunogenicity, depending on the subtypes and disease settings (early vs. metastatic BC).

Both triple-negative breast cancer (TNBC) and human epidermal growth factor-positive (HER2+) tumors are commonly enriched in tumor-infiltrating lymphocytes (TILs) (20% and 16% of cases, respectively) [6] with high immune-related gene expression [7,8]. Conversely, hormone receptor-positive (HR+) breast tumors are generally considered as the immune silent cancer type due to the absence of tumor antigens, the low expression of Major Histocompatibility Complex class I (MHC-I) molecules and the inhibition of T helper 1 (Th1) effector cells [9].

Compared to primary BC, where anticancer immune surveillance is able to destroy malignant cells, metastatic disease is characterized by the activation of several immune evasion mechanisms resulting in an inert immune environment. These findings suggest that ICIs may be more active in early-stage BC rather than in the metastatic setting. Therefore, to increase immunogenicity in metastatic BC, several combination strategies have been proposed, including anti-PD-1/PD-L1 cytotoxic drugs or other immune modulatory molecules [10,11].

To date, two ICIs, atezolizumab and pembrolizumab, have received approval from the US Food and Drug Administration (FDA) and the European Medicines Agency (EMA) as first-line treatments, in combination with chemotherapy (CT), for TNBC patients with PD-L1-positive metastatic disease based on the results of the IMpassion130 [12] and KEYNOTE-355 [13] trials, respectively. In July 2021, after the results of KEYNOTE-522, pembrolizumab was also approved by the FDA in combination with CT as a neoadjuvant treatment and, after surgery, as a single agent for nine cycles for patients with high-risk stage II or III TNBC [14].

This review provides a summary of the BC immune biology, the current knowledge on ICIs in clinical practice and an overview of the promising new strategies to enhance antitumor immune responses.

## 2. Breast Cancer “Immunogram”

### Predictors of Response to Immune-Based Therapy

The survival benefit derived from the introduction of ICIs in cancer therapy is indisputable. Unfortunately, the majority of patients experience different clinical responses to the same immunotherapy protocol with a significant proportion of treatment failures.

Therefore, one of the major challenges in immune oncology is the identification of predictive biomarkers to determine BC patients eligible for immunotherapy, so as to reduce the cost of an ineffective treatment and to avoid the risk of serious immune-related adverse events (grades 3 and 4 in approximately 5–10% of patients).

Cytotoxic T Lymphocyte Antigen 4 (CTLA-4) and programmed cell death protein 1 (PD-1) were investigated first as immune checkpoints molecules.

CTLA-4 acts as a suppressive molecule able to inhibit both the proliferation and the effector functions of T cells [15]. CTLA-4 overexpression is observed in about 50% of BC patients, but its prognostic and predictive roles remain controversial [16,17]. To date, pilot clinical trials with anti-CTLA-4 antibodies (tremelimumab and ipilimumab) in BC have yielded negative results both in terms of progression-free survival (PFS) and overall survival (OS) [18].

PD-1 is an inhibitory receptor acting as a suppressor of both adaptive and innate immune responses and is expressed on activated T-lymphocytes, particularly on tumor-specific cells, as well as on natural killer (NK) and B lymphocytes, macrophages, dendritic cells (DCs) and monocytes [19]. Programmed death ligand-1 (PD-L1) is a 40 kDa type 1 transmembrane protein expressed on human cells that is specific for the PD-1 receptor on the surface of immune effector cells. The PD-1/PD-L1 interaction plays a crucial role in maintaining self-tolerance and in the regulation of inflammation through T-cell function inhibition [20]. Not surprisingly, engagement of this ligand-receptor pair represents an adaptive immune mechanism of cancer cells to escape antitumor responses [21].

In BC, PD-L1 is more expressed in stromal immune cells (ICs) compared to tumor epithelium and is commonly associated with unfavorable clinicopathologic features (i.e., a large tumor size and poorly differentiated histological grade, high Ki67), high TIL counts, TNBC subtype and HER2+ status [22]. Despite its association with aggressive tumor characteristics, the upregulation of PD-L1 in ICs is associated with a better survival in BC [23]. Clinical trials evaluating ICIs have reported promising results in PD-L1-positive patients with advanced TNBC [12,13].

Conversely, in the neoadjuvant setting, immunotherapy seems to provide benefits regardless of the PD-L1 status [24]. Although the immune-based strategy has shown exciting therapeutic benefits for patients with PD-L1-positive metastatic TNBC (mTNBC), several studies have demonstrated that the PD-L1 status is insufficient for identifying responder patients [25]. Moreover, a recent analysis of the Tumor Cancer Genome Atlas (TCGA) revealed that PD-L1 positivity is only weakly associated with immunotherapy efficacy [26]. These discrepancies could be attributable to the lack of standardized PD-L1 assays and antibodies, as well as to the temporally and spatially heterogeneity in PD-L1 expression [27,28,29,30,31,32].

Several PD-L1 assays have been developed with different scoring systems, cutoffs and definitions to define PD-L1 positivity: SP142 (Roche Tissue Diagnostics, Tucson, AZ, USA), SP263 (Roche Tissue Diagnostics), 22C3 (Agilent Technologies Inc., Santa Clara, CA, USA) and 28-8 (Agilent Technologies Inc.).

Due to these drawbacks, the post hoc analysis of IMpassion130 assessed the analytical and clinical concordance of the DAKO 22C3, VENTANA SP142 and VENTANA SP263 assays.

The VENTANA SP142 and VENTANA SP263 assays assessed the PD-L1 expression on ICs with a 1% threshold. DAKO 22C3 evaluated the PD-L1 positivity by using a combined positive score (CPS), defined as the number of PD-L1-stained cells (tumor cells, lymphocytes and macrophages) divided by the total number of viable tumor cells multiplied by 100. These immunohistochemistry assays were not interchangeable, since the positive percentage agreements (PPA) between VENTANA SP142 IC and SP263 IC and between VENTANA SP142 IC and DAKO 22C3 CPS were 97.5% and 97.9%, respectively [33]. Recently, the VENTANA SP142 platform has been shown to display a lower sensitivity in detecting PD-L1 positivity on both tumor and immune cells with respect to DAKO 22C3 and VENTANA SP263 (46%, 81% and 75%, respectively) [33,34].

Moreover, the PD assessment may be affected by different tissue fixation/preservation methods, as inferred by higher PD-L1 scores in frozen tissues compared to their matched formalin-fixed paraffin-embedded samples [35].

Due to the spatial and temporal heterogeneity of PD-L1, it is unclear whether the evaluation should be assessed on primary or secondary lesions.

Primary BC has been reported to show higher rates of PD-L1 expression than metastatic sites, likely due to immune escape mechanisms occurring during disease progression [28,36]. Moreover, the IC assessment of metastatic sites revealed the highest prevalence of PD-L1 positivity on lymph nodes and lungs, as opposed to the liver, where an immunosuppressive microenvironment has been identified [37,38]. The PD-L1 status is also strongly modulated by treatment, with a conversion rate (from positive to negative and vice versa) ranging from 25 to 30% over time [39]. In particular, after neoadjuvant chemotherapy (NACT), PD-L1 expression has been demonstrated to be considerably expressed on a residual disease sample [40] consistent with the observations of CT inducing an adaptive immune response [41].

Since PD-L1 cannot act as a comprehensive and independent biomarker in clinical practice, several efforts have been made to identify additional biomarkers potentially able to effectively predict the treatment response to ICIs, such as the abundance of CD8^+^ TIL infiltration, tumor mutational burden (TMB), mismatch repair deficiency (dMMR), microsatellite instability (MSI) and PD-1 copy number alteration (CNA).

The clinical validity of TILs in BC is now well-established. TILs are mononuclear ICs categorized as stromal compartment TILs (sTILS) and intra-tumoral compartment TILs (iTILs), consisting of different lymphocyte subtypes, mostly T cells (cytotoxic CD8^+^ and helper CD4^+^), admixed with B cells, NK cells and macrophages [42,43].

TILs recognize neoantigens generated after cancer cell death and elicit an antitumor response through the interaction of distinct T-cell receptors (TCR) with specific neoantigen-derived epitopes presented by MHC molecules [44].

Lymphocyte predominant breast cancers (LPBC), characterized by the presence of 50–60% TILs, are associated with more favorable survival outcomes and a higher probability of a pathological complete response (pCR) after neoadjuvant therapy [45]. The prevalence of TILs is a dynamic event that is stage- and site-dependent (high or low in the case of early or advanced stages, respectively, and the variably detected depending on the sites of BC metastases, being the highest in lung metastases and lowest in liver and skin lesions) [36]. Furthermore, the production of interferon γ (IFN-γ) by activated TILs leads to PD-L1 and MHC-I upregulation, suggesting a crucial role for IFN-γ signaling in antitumor immune responses [46].

Since PD-L1 expression and TIL levels are strongly correlated with each other, the simple morphological evaluation of TILs may represent a surrogate of the activated host antitumor immune response [47].

Exploratory analyses of recent clinical trials have suggested that TILs are associated with the response to both cytotoxic and immune therapies, particularly in TNBC patients. These findings support the clinical utility of TILs in predicting the beneficial impact of immunotherapy in early and advanced TNBC settings; however, due to the retrospective nature of these data, further confirmatory independent prospective studies are needed [48].

A further promising predictor of the response to ICIs is represented by TMB, defined as the measurement of the amount of nonsynonymous mutations per coding area of a tumor genome. These mutations can be transcribed and translated, leading to the production of misfolded proteins (neoantigens) that can be recognized as non-self by T cells, thereby resulting in strong effector cell responses. Compared to other malignancies such as melanoma, lung and colorectal cancer, BC displays a lower mutation load (roughly one mutation per Mb). Indeed, a high TMB, which indicates a “hot tumor phenotype”, is found in only 3.1% of BCs and is more frequently detected in the HR-negative subtype and in older patients. These tumors, characterized by a high degree of immune infiltration, are associated with improved survival outcomes, regardless of tumor stage, molecular subtype, PD-L1 status, age and treatment schedule [49]. In BC, a high TMB is more likely associated with the MMR pathway or homologous recombination repair system deregulation, alterations in DNA polymerase genes (POLE/POLD1) and the APOBEC mutation signature [50]. Of note, compared to early BC, more advanced tumors generally display a higher TMB, probably related to the accumulation of genomic instability during disease progression or treatment-associated selective pressure, and less abundant TIL levels reflecting cancer immune escape mechanisms [48,51].

A significant correlation between a high TMB and response to ICIs has been reported in several cancers, including urothelial carcinoma, lung cancer, melanoma and human papilloma virus (HPV)-negative head and neck squamous cell carcinoma [52]. Few data are available about TMB and the response to immunotherapy in BC. In the retrospective analysis of the TAPUR (Targeted Agent and Profiling Utilization Registry) trial, the cohort of metastatic BC patients with a high TMB (defined as at least nine mutations per Mb, according to a FoundationOne test or another TAPUR-approved test) treated with pembrolizumab showed an objective response rate (ORR) and a disease control rate (DCR) of 21% and 37%, respectively [53].

Furthermore, in the KEYNOTE-119 trial, patients with previously treated mTNBC and TMB ≥ 10 mutations per Mb had tendentially a better outcome with pembrolizumab than with CT [54]. Conversely, in a large analysis including 1662 patients affected by different advanced malignancies receiving at least one dose of ICIs, no association was observed between higher TMB and improved survival in the subgroup of BC patients. Furthermore, a subgroup analysis of the GeparNuevo trial suggested an increased likelihood of pCR after NACT in the case of a high TMB independently from the addition of durvalumab [55]. These divergent results and the absence of both a well-established method of evaluation (targeted NGS panels or whole-exome sequencing) and optimal threshold to define high vs. low mutational burdens make TMB unable to predict the response to immunotherapy in BC.

During DNA replication, several errors such as the insertion, deletion and misincorporation of bases may occur. These are more frequent in non-coding short-tandem repeats in the genome known as microsatellites and are corrected by MMR proteins. When MMR genes are mutated or epigenetically silenced, they fail to repair post-DNA replicative mistakes and may lead to the MSI-high (MSI-H) phenotype characterized by alterations in the length of microsatellite regions [56]. The accumulation of mutations carried by MSI-H tumors elicits TIL immune-specific antitumor responses [57]. Therefore, MSI-H/dMMR tumors are more prone to be responsive to ICIs. Based on this evidence, pembrolizumab FDA’s approval included all solid tumors harboring this intrinsic genetic scare. The dMMR feature is extremely rare in BC, accounting for only 1 to 2% of cases. The reported low percentage of dMMR BC may be influenced by the absence of a Companion Diagnostics assay (CDx) and/or tumor-specific guidelines for a MMR analysis and the different testing methods employed, such as direct sequencing of microsatellite markers, next-generation sequencing (NGS) and immunohistochemistry (IHC) for the four MMR proteins. In BC, MMR protein loss is more commonly detected than MSI; therefore, IHC for the MMR proteins and MSI testing are not interchangeable, as in other tumor types [58]. In this context, the expression analysis of phosphatase and tensin homolog (PTEN), a key tumor suppressor involved in cell growth, proliferation and survival but also implicated in the MMR and overall DNA damage response, has been proposed to identify MMR-proficient (pMMR) breast tumors. Despite these limitations, the predictive value of MMR deficiency has been demonstrated in two reports evaluating metastatic triple-negative and luminal BC patients treated with nivolumab and pembrolizumab, respectively [59,60,61].

*BRCA1* and *BRCA2* are two suppressor genes involved in the repair of DNA double-stranded breaks. Mutations of *BRCA* genes are reported in about 5% of all diagnosed BC patients and are generally associated with increased TILs and higher PD-L1 and CTLA-4 gene expression than tumors with wild-type genes, suggesting an increased likelihood of a positive ICI response [62]. In the IMpassion130 trial, about 15% of the enrolled patients had *BRCA* mutations. In a subgroup analysis including PD-L1-positive patients, those harboring *BRCA1* or *BRCA2* mutations were shown to benefit from the immunotherapy combination more significantly than the wild-type subset. Therefore, although these genes cannot be considered independent biomarkers, they nonetheless contribute to tailoring the ICI approach [63].

During the 2020 ESMO Breast Cancer Virtual Meeting, an increase in the number of PD-L1/CD274 genes measured by CNA was proposed as a predictive marker for PD-L1 inhibitor efficacy. An exploratory translational analysis of the SAFIR-02 IMMUNO trial showed a higher efficacy of durvalumab for patients with PD-L1 copy gain (three or four copies) or amplification (>four copies) in all subtypes, as well as in TNBC [64].

Despite how PD-L1 CNA seems to be a promising biomarker, further analyses are needed to understand whether PD-L1 amplification is associated with overexpression at the protein level and the underlying biological mechanism.

## 3. Anti PD-1 Antibodies in Metastatic TNBC: Available Results from Clinical Trials

### 3.1. Pembrolizumab

Pembrolizumab is a humanized IgG4 kappa anti-PD-1 monoclonal antibody (moAb) whose effectiveness was first investigated in the KEYNOTE-012 clinical trial (NCT01848834).

KEYNOTE-012 is a multicohort phase Ib trial evaluating the efficacy and safety of single-agent pembrolizumab in PD-L1-positive patients with advanced solid tumors. In 32 heavily pretreated PD-L1-positive mTNBC, pembrolizumab administration was associated with clinical antitumor activity (ORR: 18.5%; 6-month PFS: 24.4%; 12-month OS: 43.1%) with an acceptable safety profile [65].

Following these encouraging results, the phase II KEYNOTE-086 trial (NCT02447003) tested pembrolizumab as second or later line of treatment in different cohorts of patients. In cohort A, 105 PD-L1-positive out of 170 mTNBC patients had an ORR and DCR of 5.7% and 9.5%, respectively, while the median PFS and OS were 2 and 8.8 months, respectively. These results suggest that alternative strategies, including a combination of ICIs with cytotoxic agents, should be considered in this subset of patients [66]. Conversely, in cohort B, instead, 86 PD-L1-positive mTNBC patients treated with pembrolizumab as the first line displayed an ORR of 21.4% and a median duration of the response (DoR) of 10.4 months, while the median OS and PFS were 18 and 2.1 months, respectively [67].

Subsequently, the phase III randomized KEYNOTE-119 trial (NCT02555657) compared pembrolizumab monotherapy to single-agent physician’s choice CT (capecitabine, eribulin, gemcitabine or vinorelbine) in 622 mTNBC patients. The anti-PD-1 agent, as a monotherapy in second- or third-line treatment, failed its prespecified primary endpoint of superior OS in comparison with CT [68]. However, an exploratory analysis revealed that the OS advantage registered in the experimental arm improved linearly with increasing the PD-L1 CPS. The greatest survival gain both in terms of OS and PFS was reported among patients with a CPS of 20 or higher. Notably, a great benefit was observed in women with lung (HR = 0.53; 95% CI, 0.31–0.92) or liver metastases (HR = 0.65; 95% CI, 0.31–1.38). The median DoR was not significantly different between the two arms. In terms of safety, pembrolizumab therapy is well-tolerated, with lower rates of grade 3–5 adverse events than CT, as well as toxicity, leading to a dose reduction or discontinuation [68].

The modest results registered in single agent immunotherapy trials have promoted the development of novel therapeutic strategies, including PD-1/PD-L1 inhibitors in combination with CT or targeted agents or different ICIs to enhance the immune responses through synergistic effects.

Chemotherapeutics agents can elicit antitumor immunity in different ways: by inducing immunogenic cell death, resulting in neoantigens release in the TME; by upregulating the expression of tumor antigens themselves or of MHC-I and costimulatory molecules (B7-1 and B7-2) or by downregulating checkpoint molecules (PD-L1/B7-H1 or B7-H4) expressed on the tumor cell surface, thus enhancing the strength of effector T-cell activity. Moreover, systemic therapy has also been shown to increase TILs during treatment and to block mechanisms of tumor immune evasion [69].

Based on these data, several trials with chemoimmunotherapy combinations have been conducted. The initial studies in unselected populations reported overall disappointing results [70], thus supporting the need for patient selection based on clinicopathological or molecular biomarkers able to predict those likely benefitting from the addition of ICIs.

The KEYNOTE-150/ENHANCE-1 phase Ib/II trial (NCT02513472) investigating the combination of pembrolizumab plus eribulin mesylate enrolled 107 mTNBC patients who received ≤two prior lines of systemic therapy. An ORR of 25.6% with a median PFS of 4.1 months and median OS of 16.1 months were reported. Interestingly, in the first-line setting, PD-L1-positive patients achieved a numerically doubled ORR when compared to PD-L1-negative subjects [71].

Noteworthy, in the phase III KEYNOTE-355 trial (NCT02819518), 847 mTNBC patients were randomized 2:1 to receive placebo or pembrolizumab in addition to CT (nab-paclitaxel, paclitaxel and gemcitabine-carboplatin) as the first-line treatment. Randomization was stratified by the type of on-study CT (taxane or gemcitabine–carboplatin) and PD-L1 expression at the baseline (CPS ≥ 1 or <1). The study had two coprimary endpoints, namely PFS and OS evaluated in both the intent-to-treat (ITT) population and the PD-L1-positive cohort (CPS ≥ 10 or ≥1). Secondary endpoints included ORR, DoR, DCR and safety.

A meaningful benefit from the addition of pembrolizumab was registered in the PD-L1 CPS ≥10 subgroup, in whom both the PFS and OS significantly improved. The interim analysis revealed a PFS of 9.7 months in the investigative arm vs. 5.6 months in the placebo arm (HR = 0.66; 95% CI, 0.50–0.88) [72]. The ORR and DCR were 52.7% vs. 40.8% and 65% vs. 54.4% in the two arms respectively. After a median follow-up of 44 months, the pembrolizumab-treated cohort showed a 27% reduction in the risk of death (HR = 0.73; 95% CI, 0.55–0.95; *p* = 0.0093). The median OS was 23 and 16.1 months in the experimental and control arms, respectively (HR = 0.71, 95% CI, 0.54–0.93); notably, when using a PD-L1 CPS threshold of ≥1, a benefit was observed in the PFS but not in the OS. Conversely, a similar OS advantage was reported in the subgroup with PD-L1 CPS of 10–19 (20.3 vs. 17.6 months; HR = 0.71; 95% CI, 0.46–1.09) and ≥20 (24 vs. 15.6 months, HR = 0.72; 95% CI, 0.51–1.01). Overall, these findings suggest that the observed benefit was not solely related to extremely high PD-L1 values, thus making CPS ≥ 10 a reasonable cutoff for patient selection [73].

Finally, the subgroup analysis of KEYNOTE-355, presented at the latest San Antonio Breast Cancer Symposium (2021), confirmed the improvement in survival outcomes across the patients’ subgroup, except for patients with a disease-free interval of less than 12 months. The secondary endpoints of ORR, DCR and DoR were also met in the pembrolizumab/chemotherapy arm [73].

Several clinical trials with pembrolizumab alone or in combination with other agents in mTNBC are ongoing (Table 1).

### 3.2. Nivolumab

Nivolumab is a fully human IgG4 moAb targeting PD-1. The preliminary results of its clinical activity in combination with other agents in mTNBC patients have been made available by several phase I/II trials.

The phase II WJOG9917B NEWBEAT trial (UMIN000030242), investigated the efficacy and safety of nivolumab in combination with paclitaxel and bevacizumab as a first-line therapy in 57 HER2-negative metastatic BC patients. A significant amount (83.3%) was recorded in 18 (32%) patients with TNBC [74]. Of note, a subsequent translational analysis showed a correlation between the vascular endothelial growth factor A (VEGF-A) levels and clinical outcome in this study population. In particular, the VEGF-A high subgroup had a better objective response than the VEGF-A low subset, suggesting that bevacizumab may clinically overcome the immune suppression via the inhibition of VEGF-A [75].

Encouraging results were obtained from the phase II adaptative, noncomparative TONIC trial (NCT02499367) in which 67 mTNBC patients were randomized to receive nivolumab without induction or with a 2-week low-dose induction or in addition to irradiation and CT (including cyclophosphamide, cisplatin, and doxorubicin). A higher ORR was recorded in patients treated with induction chemotherapy (23% and 35% with cisplatin and doxorubicin, respectively). The upregulation of immune-related genes involved in PD-1/PD-L1 and T-cell cytotoxicity pathways, as observed in tumor samples of patients administered chemotherapeutic induction with doxorubicin, could account for the marked antitumor activity of doxorubicin, as hypothesized by the authors [76].

Nivolumab, in combination with different chemotherapeutic agents, moAbs, ICIs, immunomodulating cytokines or vascular endothelial growth factor receptor (VEGF-R) inhibitors, is currently under evaluation in ongoing clinical trials (Table 1).

## 4. Anti PD-L1 Antibodies in Metastatic TNBC: Available Results from Clinical Trials

### 4.1. Atezolizumab

Atezolizumab is a genetically engineered, humanized IgG1 moAb that specifically binds to PD-L1, thus blocking its interaction with PD-1 [77].

Atezolizumab monotherapy was assessed in a phase I trial (NCT01375842) involving 116 mTNBC patients treated as the first or second line. Single-agent atezolizumab provided durable clinical activity and good tolerability in patients taking the first-line treatment and in those with PD-L1 expression detected in at least 1% of tumor-infiltrating ICs [78].

To extend these observations, and to exploit the immunostimulatory effect of cytotoxic drugs, a combination approach consisting of atezolizumab, and nab-paclitaxel was investigated in a phase Ib multicohort trial (NCT01633970). Thirty-three mTNBC patients, regardless of PD-L1 expression and previously treated with no more than two lines of CT in the metastatic setting, were recruited. In the whole cohort, the relevant antitumor activity in terms of the ORR, PFS and OS with an acceptable toxicity profile was reported. Interestingly, these beneficial effects were more sustained in previously untreated patients and the PD-L1-positive population [79].

The phase III IMpassion130 trial (NCT02425891) randomized in a 1:1 ratio 902 locally advanced or mTNBC patients to receive nab-paclitaxel with either atezolizumab or a placebo. The study had four prespecified coprimary endpoints using PD-L1 as the stratification parameter: (1) PFS tested in parallel in both the ITT population and (2) in patients with PD-L1-expressing ICs covering ≥1% of the tumor area (IC-positive), (3) OS-tested hierarchically in the ITT population and then, if significant, (4) in the PD-L1 IC-positive population. Compared to CT alone, the addition of atezolizumab significantly improved the PFS in both the ITT population (7.2 vs. 5.5 months; HR = 0.80; 95% CI, 0.69–0.92; *p* = 0.002) and the PD-L1-positive cohort (7.5 vs. 5.0 months; HR = 0.62; 95% CI, 0.49–0.78; *p* < 0.001). Based on these results, global health authorities approved atezolizumab plus nab-paclitaxel as a first-line treatment in patients with unresectable, locally advanced, or metastatic PD-L1-positive TNBC.

Of note, the final OS analysis failed to show a statistically significant difference between the arms in the ITT population. Therefore, due to the prespecified hierarchical design, the OS was not formally evaluated in the PD-L1-positive subgroup. However, an exploratory analysis reported an OS gain of 7.5 months in favor of the atezolizumab plus nab-paclitaxel combination restricted to patients with PD-L1-positive disease, regardless of post-progression therapies. These findings support the predictive value of PD-L1, while the *BRCA* mutational status does not seem to affect the clinical benefits derived from this combination strategy [80].

Conversely, the IMpassion131 trial (NCT03125902), which evaluated the first line atezolizumab/placebo plus paclitaxel in mTNBC patients, failed to reach the threshold for statistical significance for both PFS and OS. Several hypotheses have been made to explain these unexpected and conflicting results. The different chemotherapeutic backbone (nab-paclitaxel or paclitaxel) could have resulted in a distinct response related to a specific immunogenic effect played by the single cytotoxic drug on the TME. However, this hypothesis was refuted by the results of the KEYNOTE-522 study, in which the beneficial effect of pembrolizumab was independent from the CT backbone. Another possible reason could be represented by the immunosuppressive effect of corticosteroids widely used as comedication in IMpassion131 to prevent hypersensitivity reactions and as supportive therapy. Furthermore, slight differences in the study populations can be identified. For example, the IMpassion131 trial included a higher percentage of Asian ethnicity and a lower number of patients with de novo metastatic disease, as well as more patients with one to three metastatic sites compared to IMpassion130. All these differences in the clinical features could have influenced the results [81].

### 4.2. Avelumab

Avelumab is a human IgG1 moAb directed against PD-L1. Moreover, unlike other PD-L1 blocking moAbs, avelumab can potentially mediate antibody-dependent cell cytotoxicity (ADCC) against tumor cells. In vitro experiments have shown that avelumab triggered ADCC against TNBC cells expressing detectable levels of PD-L1, with a significant increase in tumor cell lysis independently of the blockade of the PD-1/PD-L1 axis; besides its use with immunomodulators such as interleukin (IL) 2 or IL-15 may improve the therapeutic efficacy of avelumab itself [82].

In the phase I JAVELIN trial (NCT01772004), single agent avelumab was given to 168 pretreated metastatic BC patients, including 58 participants with TNBC. In the total cohort and, also, in the TNBC subgroup as well, the ORR was very modest (3% and 5.2%, respectively). As for pembrolizumab and atezolizumab, a stronger antitumor effect was reported in PD-L1-positive patients compared to the PD-L1-negative cohort, both in the overall population (16.7% vs. 1.6%) and in the TNBC subgroup (22.2% vs. 2.6%). The treatment was safe, with grade ≥3 adverse events occurring in 13.7% of patients, including two treatment-related deaths [83].

Avelumab alone or in a combination strategy is under investigation in the clinical trials indicated in Table 1.

### 4.3. Durvalumab

Durvalumab is an engineered human IgG1 moAb that binds with high affinity and specificity to PD-L1 [84]. In the phase II SAFIR02-BREAST IMMUNO trial (NCT02299999), 199 patients with HER2-negative metastatic BC, who did not progress after six to eight cycles of induction CT and lacking a targetable molecular alteration, were randomized 2:1 to durvalumab or CT (paclitaxel, capecitabine and fluorouracil/epirubicin/cyclophosphamide) as the maintenance therapy. In patients with TNBC (*n* = 82), durvalumab, compared to CT, significantly improved the OS (21.2 vs. 14 months, HR = 0.54; 95% CI, 0.30–0.97, *p* = 0.037). Interestingly, an exploratory analysis showed a significant advantage for patients with PD-L1^-^positive TNBC and for those with CD274 gain/amplification, while tumor infiltration by lymphocytes (CD8, FoxP3 and CD103 expressions) and homologous recombination deficiency were not associated with sensitivity to durvalumab [64].

Despite the paucity of participants (n = 24), encouraging a results have been registered in a small phase I/II single arm trial (NCT02628132) evaluating the safety and the efficacy of durvalumab in combination with paclitaxel (5 and 20.7 months for the PFS and OS, respectively) [85].

BEGONIA is a phase Ib/II multi-arm trial (NCT03742102) testing durvalumab with or without paclitaxel in combination with novel anticancer drugs (capivasertib, oleclumab, trastuzumab deruxtecan, datopotamab and deruxtecan) as first-line treatments for mTNBC. The initial results from arm 1, durvalumab plus paclitaxel, revealed an ORR of 57% and PFS of 7.3 months [86].

## 5. Anti PD-1 Antibodies in Early TNBC: Available Results from Clinical Trials

### Pembrolizumab

The use of pembrolizumab combined with CT was investigated in several clinical trials in the neoadjuvant setting, resulting in a significant improvement of pCR in TNBC patients.

The I-SPY2 (Investigation of Serial Studies to Predict Your Therapeutic Response With Imaging and Molecular Analysis 2) is an ongoing multi-center, phase II, randomized trial (NCT01042379) testing anthracycline- and taxane-based NACT with or without pembrolizumab in patients with high-risk (by MammaPrint), stage II/III, HER2-negative BC. The preliminary results are available, showing increased pCR rates in pembrolizumab plus the standard NACT vs. CT alone, with a remarkable efficacy in the TNBC phenotype, where a three-fold higher response rate was registered (22% vs. 60%, control vs. investigative arm, respectively). Furthermore, the achievement of pCR was clearly associated with a substantially improved outcome in term of event-free survival (EFS) rates (93% at a 3-year median follow-up) [87].

Subsequently, two additional studies confirmed the clinical benefit of pembrolizumab combined with CT in the preoperative scenario.

KEYNOTE-173 is a multi-cohort phase Ib study (NCT02622074) including high-risk, locally advanced TNBC designed to assess the efficacy and safety of pembrolizumab plus six NACT regimens (paclitaxel, nab-paclitaxel, doxorubicin, cyclophosphamide, and carboplatin) administered at different times and with distinct dosing schedules. The pCR rate across all cohorts was 60% (range 49–71%). Exploratory biomarker analyses showed that higher TIL counts and PD-L1 positivity were significantly associated with a higher likelihood of pCR. Moreover, the achievement of pCR was shown to better predict long-term survival outcomes both in terms of the EFS and OS rates [88].

During the ESMO 2021 Virtual Plenary session, the fourth interim analysis of the KEYNOTE-522 study (NCT03036488) was presented. In this trial, 1174 patients with newly diagnosed TNBC were randomized 2:1 into four cycles of pembrolizumab or placebo with paclitaxel plus carboplatin followed by doxorubicin/epirubicin and cyclophosphamide and nine cycles of pembrolizumab or placebo after surgery. Compared to CT alone, the addition of pembrolizumab in the perioperative strategy demonstrated an absolute pCR improvement of 13.6% (64.8% vs. 51.2%), as well as a significant and clinically meaningful reduction in EFS events (HR = 0.63; 95% CI, 0.48–0.82; *p =* 0.00031) regardless of PD-L1 levels [89]. The investigators performed a subgroup analysis showing that all participants seemed to obtain a comparable EFS advantage irrespective of the nodal involvement, overall disease stage, menopausal status, HER2 expression and lactate dehydrogenase levels [90].

Following these promising results, a wide plethora of trials testing pembrolizumab in the early stage have been designed and are currently ongoing (Table 2).

## 6. Anti PD-L1 Antibodies in Early TNBC: Available Results from Clinical Trials

### 6.1. Atezolizumab

In the phase III IMpassion031 trial (NCT03197935), 133 patients with previously untreated stage II or III diseases were randomly assigned to either atezolizumab or a placebo plus CT, which consisted of weekly nab-paclitaxel followed by biweekly doxorubicin and cyclophosphamide as a neoadjuvant treatment. The experimental arm also received 1 year of adjuvant atezolizumab. Like the abovementioned KEYNOTE-522, this study met its primary endpoint showing in the atezolizumab arm a statistically significant advantage in terms of pCR (57.6% vs. 41.1%, *p* = 0.0044). Although the median EFS was not reached in both arms at the time of analysis, the EFS events were numerically lower in the atezolizumab arm (10.3% vs. 13.1% in the CT-alone arm) [91].

Opposite results arose from the NeoTRIP/Michelangelo phase III trial (NCT02620280), which evaluated the addition of atezolizumab or placebo to neoadjuvant carboplatin/nab-paclitaxel, followed by surgery and adjuvant anthracycline-based regimen in women with early high-risk or locally advanced, unilateral TNBC. Specifically, adding atezolizumab did not result in a statistically significant difference in the pCR rate vs. CT alone either in the ITT (43.5% vs. 40.8%) or PD-L1-positive subset (51.9% vs. 48%) [92].

This discrepant data led to important considerations and additional translational analyses to explain these unexpected results. To this end, the investigators assessed the expression and dynamics of sTILs and iTILs and their association with pCR. These exploratory analyses revealed that the combination of anti-PD-L1 and NACT increased pCR by 10% or more in “immune-rich” tumors (PD-L1 IC-positive, high sTILs/iTILs). In particular, a positive relationship between the PD-L1 IC expression levels and pCR rates in the overall cohort and in each treatment arm was registered (52.3% with atezolizumab and 47.7% with CT). Additionally, an evaluation of the tumor samples revealed that, while the PD-L1 IC levels were balanced across all groups, both baseline sTILs and iTILs were disproportionately higher in the control arm, which could be responsible for the small differences in pCR seen between the groups. Furthermore, in both arms, most patients demonstrated an increase of the TILs after one cycle, suggesting a key role of CT in TME shaping [93]. Therefore, although the baseline PD-L1 and TILs status were not found to be predictive of the ICI efficacy in the neoadjuvant setting, they seem to have a role as predictors of CT benefits.

It should be noted that, in contrast to KEYNOTE-522 and IMpassion031, NeoTRIP included more than half of the participants with high tumor burden (stage III disease), which is well-known to be associated with more impaired antitumor immune responses, with resultant blunted treatment responses. Additionally, the omission of anthracycline in NACT backbone could have contributed to a lower response rate due to its ability to promote antitumor immunogenicity by increasing the percentage of PD-L1^-^positive BC cells, as demonstrated in preclinical models [94]. Regarding EFS, the data are still inconclusive, thus, despite the lack of a significant pCR improvement, a survival advantage from adding immunotherapy to NACT could not be excluded.

The utility of atezolizumab in the neoadjuvant setting is currently being investigated in the GeparDouze/NSABP B-59 trial (NCT03281954) in which TNBC patients were randomized to neoadjuvant atezolizumab or placebo combined with CT (carboplatin plus paclitaxel and epirubicin/adriamicin plus cyclophosphamide), followed by surgery and atezolizumab or placebo as adjuvant therapy to complete 1 year of treatment.

The phase III ALEXANDRA/IMpassion030 trial (NCT03498716) aims to assess whether adding atezolizumab to the standard CT after surgery is better suited for preventing cancer recurrence compared to chemotherapy alone in TNBC patients.

### 6.2. Durvalumab

The GeparNuevo study (NCT02685059) aimed at investigating durvalumab or placebo, administered with concomitant weekly nab-paclitaxel followed by standard CT, with epirubicin/cyclophosphamide in patients with early TNBC. The study included an initial 2-week window of opportunity (WOP) during which patients received durvalumab or a placebo monotherapy prior to beginning CT for the first 117 patients enrolled. The primary analysis showed little improvement in the pCR rate with durvalumab vs. the placebo (53.4% vs. 44.2%). The advantage of durvalumab was statistically significant only in the WOP cohort, with a pCR rate of 61% vs. 41.4% of the placebo group (*p* = 0.035) [95]. In both arms, the pCR rate was demonstrated to be linearly associated with the TIL levels, TMB and immune gene expression profile (GEP) [96]. Despite a modest improvement in the pCR rate (Δ of 9%), achieving a pCR corresponds to a significant advantage in the long-term outcomes (invasive DFS, distant DFS and OS) in the durvalumab cohort [97].

Durvalumab, in combination with other cytotoxic drugs or a molecular target in the neoadjuvant setting, is under evaluation in several ongoing phase I/II trials (Table 2).

## 7. Strategies to Overcome Immune Resistance

The approval of ICIs has added a new weapon in the therapeutic armory against TNBC. Despite the encouraging results, some patients do not respond to the initial immunotherapy (primary resistance), while others, after a period of treatment benefit, develop an acquired resistance. Immunotherapy resistance is multifaceted and can be driven by either tumor cell intrinsic factors or extrinsic causes that involve the TME [1].

The major tumor-induced mechanisms of immune resistance include: (1) variations in the gene expression (i.e., anti-PD-1 resistance signature [IPRES]), (2) alterations in the antigen processing pathway (i.e., IFN-γ), (3) reduction or loss of tumor antigens expression, (4) loss of MHC expression, (5) deregulation of the signaling pathways (i.e., WNT, PI3K/AKT/mTOR and MAPK); (6) release of immune suppressive cytokines (C-C Motif Chemokine Ligand 2 (CCL2), VEGF, IL-8 and Transforming Growth Factor Beta (TGF-β)) and the (7) metabolic reprogramming of cancer cells, namely the “*Warburg effect*” for the adaptation to a hypoxic environment [98,99,100,101].

Many extrinsic factors operating within the TME can affect the outcome of immunotherapy. During immunotherapy, many alterations in the composition and functionality of TME cells have been described, including: (1) the migration to TME of such immunosuppressive cells as Tregs, myeloid-derived suppressor cells (MDSCs), Tumor-associated macrophages (TAM), particularly M2 macrophages and protumor N2 neutrophils; (2) PD-L1 the upregulation on tumor cells, matched by an expression of other inhibitory receptors such as CTLA-4, T-cell immunoglobulin 3 (TIM-3), Lymphocyte activation gene-3 (LAG-3) and the inducible T-cell co-stimulator (ICOS)/CD278 on the surface of immunosuppressive cells; (3) secretion of immunosuppressive factors such as VEGF, TGF-β and and IL-10. All these factors induce a severe inhibition and exhaustion of effector T cells. T cell exhaustion is associated with impaired T cell function resulting in the development of resistance to ICIs [98,99,102,103,104].

The growing knowledge of immune resistance mechanisms sets the stage for the optimization of the current strategies and encourages trials for alternative therapeutic approaches aimed at re-sensitizing resistant tumors or at improving the effectiveness of immunotherapy. In order to overcome the resistance to ICIs, novel combinations integrating ICIs with different agents, such as targeted agents or new-generation immune modulators have been conceived. Vaccines with tumor-associated antigens, oncolytic virotherapy and adoptive cell transfer (ACT) therapy, entailing the use of TILs or CAR-T cells, may be promising novel therapeutic strategies.

## 8. Combining Immunotherapy and Targeted Therapy

### 8.1. ICIs and PARP Inhibitors

Immunoediting is a consequence of a T cell-dependent immunoselection process leading to the outgrowth of cancer subclones lacking neoantigens expression, thus conferring poor immunogenicity and resistance to ICIs [105,106].

On the contrary, BCs carrying *BRCA 1/2* mutations display higher immunogenicity due to the accumulation of nonsynonymous mutations resulting in tumor neoantigen expression [107,108] and marked TIL infiltration in the TME [109]. Likewise, DNA-damaging agents are known to be able to make tumors more immune-responsive by promoting neoantigens release, increasing TMB and enhancing PD-L1 expression. PARP inhibitors, drugs that block the single-stranded DNA repair process causing DNA damage, stimulate the release of type I interferons and other proinflammatory cytokines involved in DC maturation through the activation of the DNA-sensing cGMP synthase stimulator of interferon genes (cGAS-STING) pathway and, in turn, trigger antitumor T-cell immunity [110,111]. These insights underpin the scientific rationale for the synergistic combination of PARP inhibitors with anti-PD-1/PD-L1 agents.

Durvalumab with olaparib was tested in the phase I/II MEDIOLA basket trial (NCT02734004), in patients carrying a germline *BRCA* mutation with various advanced solid tumors, including 34 HER2-negative metastatic BC women (16 HR+, 18 TNBC) pretreated with up to two prior lines of CT. In the TNBC cohort, two subgroups with different outcomes were identified: one group of patients having an early disease progression and another one with a durable response to treatment (median DoR 12.9 months) [112]. Of note, the rate of non-responder patients (5/18) could have affected the median PFS, which was only 4.9 months in the TNBC subset, while the OS was 20.5 months.

The TOPACIO/KEYNOTE-162 trial (NCT02657889) showed promising data from the use of pembrolizumab in combination with niraparib in 47 TNBC patients previously treated with at least one prior line of therapy for advanced disease. The ORR and DCR were 21% and 49%, respectively, regardless of the *BRCA1/2* or PD-L1 status. However, at the biomarker-defined analysis, patients with tumor *BRCA* mutations (*n* = 15) or with PD-L1-positive disease (*n* = 28) had a numerically higher ORR compared to those with *BRCA* wild-type (ORR = 47% vs. 11%) or PD-L1-negative BC (ORR = 32% vs. 8%) [113].

In early stage disease, the I-SPY2 trial (NCT01042379) compared durvalumab plus olaparib and paclitaxel to paclitaxel alone as the preoperatory treatment for HER2-negative BC patients. After surgery, all participants received doxorubicin and cyclophosphamide. In the TNBC subgroup, the pCR rate was 47% in the combinatorial approach arm and 27% in the control arm with paclitaxel alone. An exploratory analysis revealed a positive correlation between the pCR and low CD3/CD8 gene signature ratio, high macrophage/T-cell MHC-II signature ratio and high proliferation signature [114]. Therefore, many studies are currently underway to further explore the role of PARP inhibitors as immune modulatory molecules in combination with ICIs as an induction therapy.

NCT03594396 is an ongoing phase II WOP trial exploring the biological effect of durvalumab plus olaparib administered before the standard NACT in early TNBC patients. Moreover, the phase II PHOENIX DDR/anti PD-L1 study (NCT03740893) is now evaluating the additive effect of one cycle of durvalumab, olaparib or an ataxia telangiectasia and Rad3-related protein (ATR), a DNA repair protein, inhibitor in TNBC patients with a high residual disease burden, as defined by MRI, after NACT.

### 8.2. ICIs and Small Molecules

The deregulation of signaling pathways, due to aberrations in oncogenes and tumor suppressors genes, influence the TME by changing the immune cell composition and cytokine profile, making tumors resistant to ICIs [115].

The loss of *PTEN* results in a constitutive activation of the PI3K/Akt/PTEN pathway, conferring a more aggressive cancer behavior and drug resistance [116]. Moreover, recent evidence has demonstrated a critical role in T-cell functions, including proliferation, survival, migration and metabolism [117,118]. Tumor cells with mutations in the PI3K/Akt/PTEN pathway tend to be less immunogenic and develop resistance to anti-PD-1/PD-L1 therapy due to the release of anti-inflammatory cytokines, such as CCL2 and VEGF, depletion of cytotoxic T cells in tumors and decreased expression of IFN-γ and granzyme B [98].

The preclinical study showed that treatment with a selective PI3Kβ small molecule inhibitor, GSK2636771, improved the efficacy of both anti-PD-1 and anti-CTLA-4 antibodies by enhancing the expansion of tumor-specific lymphocytes [119].

Similarly, the hyperactivation of MAPK signaling (also known as the Ras-Raf-MEK-ERK pathway) in BC is also involved in immunotherapy resistance. Genomic or transcriptomic activation of MAPK signaling impairs the recruitment and function of TILs through the expression of VEGF and multiple other inhibitory cytokines and suppresses the expression of MHC-I and MHC-II [120]. The preclinical models have demonstrated that MEK inhibitor treatment induces immunogenic cell death, increases the levels of effector CD8+ T cells [121] and restores the surface expression of MHC-I and PD-L1 through STAT activation, thus enhancing tumor immunogenicity [122].

Several studies have been designed to investigate the synergistic effect of ICIs with MEK and AKT inhibitors in both early and metastatic settings (Table 1 and Table 2).

VEGF is a key driver for angiogenesis and then for tumor growth. The resulting abnormal vasculature decreases the migration and activation of effector T cells and promotes the expansion of Tregs and MDSCs in the TME, leading to an immune silent tumor profile [123]. Antiangiogenic therapy is able to shape the TME, promoting tumor vascular normalization, blood perfusion, immune cell recruitment and DC maturation [124].

Despite the slight efficacy demonstrated by antiangiogenic agents in BC, many trials are exploring whether bevacizumab or VEGF-R inhibitors in combination with immunotherapy could improve clinical outcomes in patients with metastatic disease (Table 1).

### 8.3. ICIs and Novel Immune-Modulators

The overexpression of alternate immune checkpoint receptors is recognized as a potential cause of resistance opening the way for clinical trials testing novel ICIs against LAG-3, TIGIT, TIM-3, VISTA and ICOS combined with traditional PD-1 or PD-L1 inhibitors (Table 1).

For example, LAG-3 is an inhibitory receptor expressed on activated CD4^+^ and CD8^+^ T cells, a subset of Tregs, NK cells, B cells and plasmacytoid DCs showing direct suppressive activity on T-cell activation, proliferation, and homeostasis and indirect suppressive function through Treg. Thus, LAG-3 may be a potential target to enhance the anticancer immune response, with a possible synergistic activity resulting from simultaneous inhibition of the PD-1/PD-L1 pathway [125,126]. In a multicohort phase I/II trial (NCT03499899), ieramilimab (LAG525, a monoclonal antibody blocking binding of LAG-3 to MHC-II) plus spartalizumab (a monoclonal antibody blocking the interaction of PD-1 with PD-L1 and PD-L2) produced durable responses in women with mTNBC associated with a switch from an immune-cold to an immune-activated biomarker profile in tumor biopsies [127].

Other promising strategies are represented by immunomodulating agents or cytokines agonists in combination with ICIs. In this scenario, Imprime PGG is a novel immune agonist that acts as pathogen-associated molecular patterns (PAMPs) directly activating innate immune effector cells, thus triggering a coordinated anticancer immune response [128]. The results of the phase II IMPRIME1 trial (NCT02981303) reported positive data in terms of the ORR, PFS and OS from the combination of pembrolizumab and Imprime PGG in the second line or later for mTNBC patients. Moreover, translational research showed the activation of both innate and adaptive immunity with the combination regimen. These findings strongly encourage further larger controlled studies to confirm the advantages of this novel therapy [129]. Finally, Bempegaldesleukin (BEMPEG/NKTR-214) is an immunostimulatory IL-2 cytokine prodrug that acts as the IL-2 pathway signal agonist. BEMPEG works by stimulating the proliferation of specific cancer-killing T and NK cells without expanding intratumoral T cells, which dampen the immune response.

In the phase II multicohort PIVOT-02 trial (NCT02983045), the combination of the BEMPEG and nivolumab in 43 mTNBC patients was associated with promising preliminary clinical activity in terms of the response and safety data [130].

### 8.4. ICIs and Epigenetic Agents

Epigenetic modification may contribute to primary and acquired resistance during ICIs therapy because of interference in many aspects of antitumor immunity: neoantigen presentation and processing; T-cell functions, differentiation, and proliferation; memory T-cell phenotype acquisition; T-cell migration and T-cell exhaustion. Epigenetic targeting agents have displayed antitumor activity either as a monotherapy or in combination with immunotherapy in preclinical studies. Histone deacetylase inhibitors (HDACi) by the epigenetic regulation of gene expression are able to induce cancer cell cycle arrest and death, hindering angiogenesis and the modulation of immune responses. In vitro experiments have shown that HDACi upregulate PD-L1 expression on tumor cells while reducing Tregs in the TME [131]. Despite the promising findings from these preclinical data, the results of the phase II ENCORE 602 (TRIO025) trial (NCT02708680) failed to demonstrate a clinical advantage with the addition of entinostat (a selective class I HDACi) to atezolizumab in pretreated mTNBC patients [132].

## 9. Adoptive Cell Therapy (ACT)

Many technologies are emerging with the aim of shaping the composition and functionality of TME cells. One method is represented by ACT, a strategy consisting of the transfer of autologous or allogenic TILs or T cells genetically engineered to express modified TCR or chimeric antigen receptors (CAR).

In particular, adoptive TIL therapy is based on the isolation of TILs from the TME, ex vivo activation and expansion by using high doses of IL-2 and, finally, reinfusion back into the patient [133]. This method has been tested for all BC subtypes in several early phase trials (NCT04111510, NCT01462903 and NCT01174121).

Unfortunately, not all TIL-derived T cells are tumor-responsive; some are characterized by a low survival, and some can be difficult to activate and expand after reinfusion into the patient’s blood. To overcome these obstacles, another technique has been employed in the ACT, which entails the use of activated T cells, as well as NK cells, obtained from the patient and then genetically modified by means of viral vectors or other nonviral gene delivery methods to express synthetic TCR or CAR that enables them to target specific cancer antigens [134]. After reinfusion in the patient’s circulation, the engineered TCRs exert cancer cell recognition through MHC protein binding, while CAR-T lymphocytes recognize tumor-associated antigens (TAAs) and trigger a cytotoxic immune response in an MHC-independent manner [135].

Many targets have been identified for CAR-T-cell therapy for BC such as mesothelin, epidermal growth factor receptor (EGFR), intercellular adhesion molecule-1 (ICAM), Mucin1 glycoprotein (MUC1), receptor-tyrosine-kinase-like orphan receptor 1 (ROR1) and tumor endothelial marker (TEM8). The preliminary interesting results are already available (NCT02414269) [136,137,138].

## 10. Vaccines and Oncolytic Virotherapy

Cancer vaccines represent a therapeutic approach to restore the patient’s T-cell response to TAA or tumor-specific antigens (TSAs). While TAAs originate as self-antigens, limiting the efficacy of T-cell responses due to the potential self-tolerance mechanisms, TSAs derived from somatic mutations in individual cancer cells exhibit a strong affinity toward human leukocyte antigen (HLA)/MHC and TCR. TSAs, being fully cancer-specific, are able to bypass the central tolerance, thus representing ideal candidates for personalized immunotherapy. Cancer vaccines may activate an antitumor immune response to prevent or treat cancer. There are several vaccination strategies, based on different immunization modalities: cancer antigens, through peptide, protein or engineered plasmid DNA; cells, such as DCs or autologous tumor cells; tumor cell lysates, derived from individual patients.

In BC, vaccine monotherapy has been associated with a modest immune response and limited antitumor activity. Given the low immunogenicity and heterogeneity of BC, various approaches have been explored to raise the efficacy of cancer vaccines.

For instance, the combination of PD-1/PD-L1 inhibitors with DC-based vaccines has been shown to produce measurable antitumor activity and survival benefits in mice models [139].

Furthermore, the results from many preclinical studies suggest that the efficacy of immunotherapies in BC can be greatly enhanced by combinations with such treatments as oncolytic virus therapy, which can favorably modulate the tumor immune landscape. Oncolytic virotherapy is a form of immunotherapy exploiting RNA- or DNA-attenuated viruses designed to selectively infect and destroy tumor cells while sparing normal cells. Oncolytic virus-mediated oncolysis causes the release of danger signals, namely damage-associated molecular patterns (DAMPs) and PAMPs, as well as TSA or TAA antigens, prompting DCs to generate a tumor-specific adaptive immune response [140]. Preclinical models have shown a synergistic cytotoxic activity when combining pelareorep, a serotype 3 reovirus, with microtubule targeting agents. Subsequently, a randomized phase II study (NCT01656538) of weekly paclitaxel with or without pelareorep in patients with metastatic BC demonstrated a significantly longer OS in the combination arm [141]. Moreover, in the vitro study, they have demonstrated that the reovirus was able to induce an immune response against BC cells when combined with anti-PD-1 therapy [142]. A phase II trial (NCT04215146) has been designed to confirm these data by combining pelareorep and paclitaxel with avelumab. Oncolytic virus-based strategies are in very early phase development and are currently under investigation in several recruiting studies (NCT04102618, NCT04301011 and NCT04185311).

## 11. Conclusions

Immunotherapy has added new therapeutic options for selected patients with TNBC. Although effective in many other tumor types, the use of ICIs as a monotherapy has shown moderate success only in a small subset of TNBC patients; so far, several combination strategies are emerging as new potential opportunities to enhance immune responses to tumors. To date, a significant clinical advantage is observed when immunotherapy is combined with conventional CT as the first-line treatment for PD-L1-positive mTNBC. However, despite the satisfying results, primary or acquired treatment resistance still occurs. The TME, in which tumor and immune cells interact, has emerged as a key player in determining therapeutic outcomes. Unfortunately, the driving resistance mechanisms to ICIs and the ways to counteract them are not fully understood. Therefore, the identification of effective prognostic and predictive biomarkers of a response currently represent a top priority in clinical practice to improve the patients’ selection. Moreover, the identification of novel targets for specific combinations or innovative approaches to overcome the limitations of the current cancer immunotherapies may help avoid ineffective treatments and limit unnecessary toxicity.

## Figures and Tables

**Figure 1 cancers-14-02102-f001:**
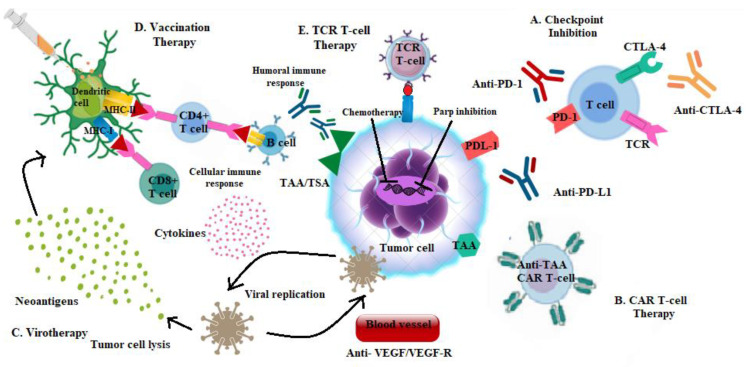
Immunotherapeutic strategies for the treatment of TNBC. Several studies are combining checkpoint blockades with multiple therapies, including traditional chemotherapy; PARP inhibitors; anti-VEGF/VEGF-R agents; anti-CTLA-4 antibodies or novel strategies including virotherapy, vaccination therapy, CAR T-cell or TCR T-cell therapy in order to overcome the mechanisms involved in the impairment of antitumor immune responses. Abbreviations: TCR: T-cell receptor; MHC-I: Major histocompatibility complex class I; MHC-II: Major histocompatibility complex class II; CTLA-4: Cytotoxic T-Lymphocyte Antigen 4; PD-1: Programmed cell death protein 1; PD-L1: Programmed death ligand 1; CAR: Chimeric antigen receptor; TAA: Tumor-associated antigen; TSA: Tumor-specific antigen; TCR:T cell receptor; VEGF: vascular endothelial growth factor; VEGF-R-: vascular endothelial growth factor receptor.

**Table 1 cancers-14-02102-t001:** List of completed or ongoing clinical trials with immune checkpoint inhibitors alone or with combinational drugs and novel immune-modulating strategies for mTNBC treatment.

NCT	Anti-PD-1/PD-L1	Non IT Drugs	IT Drugs	Comparator Arms	Experimental Arms	Phase	Primary Endpoint	Status
NCT03424005	Atezolizumab	Capecitabine Atezolizumab Ipatasertib SGN-LIV1A Bevacizumab Nab-Paclitaxel Sacituzumab Govitecan Gemcitabine Carboplatine	Tocilizumab Selicrelumab	Atezolizumab + Nab-Paclitaxel Capecitabine	Atezolizumab + Nab-Paclitaxel + Tocilizumab Atezolizumab + Sacituzumab Govitecan Atezolizumab + Ipatasertib Atezolizumab + SGN-LIV1A Atezolizumab + Selicrelumab + Bevacizumab Atezolizumab + CT	I/II	ORR Safety	Recruiting
NCT02849496	Atezolizumab	Olaparib			Olaparib Olaparib + Atezolizumab	III	PFS	Recruiting
NCT03202316	Atezolizumab	Cobimetinib Eribulin			Atezolizumab + Cobimetinib, + Eribulin Atezolizumab + Eribulin	II	ORR	Recruiting
NCT02425891	Atezolizumab	Nab-paclitaxel		Placebo Plus Nab-Paclitaxel	Atezolizumab + Nab-Paclitaxel	III	PFS in ITT PFS in PD-L1 + OS in ITT OS in PD-L1+	Completed
NCT04177108	Atezolizumab	Ipatasertib Paclitaxel		Cohort 1: PD-L1- Paclitaxel+ Placebo+ Placebo Cohort 2: PD-L1+ Paclitaxel + Atezolizumab + Placebo	Cohort 1: PD-L1-Paclitaxel + Atezolizumab + Ipatasertib Paclitaxel + Ipatasertib + Placebo Cohort 2: PD-L1+ Paclitaxel + Atezolizumab + Ipatasertib	III	PFS OS	Active, not recruiting
NCT03961698	Atezolizumab	Eganelisib Nab-Paclitaxel Bevacizumab			Cohort 1: PD-L1+ Eganelisib + Nab-Paclitaxel + Atezolizumab Cohort 2: PD-L1+ Eganelisib + Nab-Paclitaxel + Atezolizumab	II	CRR	Recruiting
NCT04408118	Atezolizumab	Paclitaxel Bevacizumab			Atezolizumab + Paclitaxel + Bevacizumab	II	PFS	Recruiting
NCT02322814	Atezolizumab	Cobimetinib Paclitaxel Nab-Paclitaxel		Cohort 1: Placebo + Paclitaxel	Cohort 1: Cobimetinib + Paclitaxel Cohort 2: Cobimetinib + Paclitaxel + Atezolizumab Cohort 3: Cobimetinib + Nab-Paclitaxel + Atezolizumab	II	Cohort 1: PFS Cohort 2, 3: OR, CR, PR	Completed
NCT03829501	Atezolizumab		KY1044 (Alomfilimab)		KY1044 KY1044 + Aezolizumab	I/II	Safety Tolerability ORR DLTs	Recruiting
NCT03101280	Atezolizumab	Rucaparib			Rucaparib + Atezolizumab	I	Safety DLTs RP2D	Completed
NCT03915678	Atezolizumab		BDB001		Atezolizumab + BDB001+ RT	II	Activity measured in terms of CR, PR, SD	Recruiting
NCT04639245	Atezolizumab	Cyclophosphamide Fludarabine	MAGE-A1-specific T Cell Receptor-transduced Autologous T-cells		FH-MagIC TCR-T cells + Atezolizumab After lymphodepletion with cyclophosphamide + fludarabine	I/II	Safety ORR	Recruiting
NCT02708680	Atezolizumab		Entinostat	Atezolizumab+ Placebo	Atezolizumab + Entinostat	II	DLT MTD PFS	Active, not recruiting
NCT02819518	Pembrolizumab	Nab-paclitaxel Paclitaxel Gemcitabine Carboplatin	Placebo + CT		Pembrolizumab + Nab-paclitaxel Pembrolizumab + Paclitaxel Pembrolizumab + Gemcitabine/Carboplatin	III	Safety PFS in ITT PFS in PD-L1 CPS ≥1 Tumors PFS in PD-L1 CPS ≥10 Tumors OS in ITT OS in PD-L1 CPS ≥1 Tumors OS in PD-L1 CPS ≥10 Tumors	Active, not recruiting
NCT02971761	Pembrolizumab	Enobosarm			Pembrolizumab + Enobosarm	II	Safety CBR DLTs	Active, not recruiting
NCT02657889	Pembrolizumab	Niraparib			Pembrolizumab + Niraparib	I/II	ORR DLTs	Completed
NCT03106415	Pembrolizumab	Binimetinib			Pembrolizumab + Binimetinib	I/II	ORR MTD	Active, not recruiting
NCT03797326	Pembrolizumab	Lenvatinib			Pembrolizumab + Lenvatinib Lenvatinib	II	ORR Safety	Active, not recruiting
NCT03184558	Pembrolizumab	Bemcetinib			Pembrolizumab + Bemcetinib	II	ORR	
NCT03272334	Pembrolizumab	HER2Bi armed activated T-cells			Pembrolizumab + HER2Bi armed activated T-cells	I/II	DLTs	Recruiting
NCT03012230	Pembrolizumab	Ruxolitinib			Pembrolizumab + Ruxolitinib phosphate	I	MTD Safety	Recruiting
NCT04468061	Pembrolizumab	Sacituzumab Govitecan			Pembrolizumab + Sacituzumab govitecan Sacituzumab govitecan	II	PFS	Recruiting
NCT04683679	Pembrolizumab	Olaparib			Pembrolizumab + Olaparib + RT Pembrolizumab + RT	II	ORR	Recruiting
NCT02411656	Pembrolizumab				Pembrolizumab	II	DCR	Recruiting
NCT02981303	Pembrolizumab		Imprime PGG		Imprime PGG + Pembrolizumab	II	ORR	Completed
NCT03650894	Nivolumab	Bicalutamide	Ipilimumab		Nivolumab + Ipilimumab, + Bicalutamide	II	CBR	Recruiting
NCT03098550	Nivolumab		Daratumumab		Nivolumab + Daratumumab Nivolumab	I/II	Safety	Completed
NCT04159818	Nivolumab	Cisplatin Doxorubicin			Nivolumab Nivolumab + Cisplatin as induction therapy Nivolumab + Doxorubicin as induction therapy	II	PFS	Recruiting
NCT02393794	Nivolumab	Romidepsin Cisplatin			Romidepsin + Cisplatin Romidepsin + Cisplatin + Nivolumab	I/II	RP2D ORR	Active not recruiting
NCT03414684	Nivolumab	Carboplatin			Nivolumab + Carboplatin Carboplatin	II	PFS	Active not recruiting
NCT03316586	Nivolumab	Cabozantinib			Nivolumab + Cabozantinib	II	ORR	Completed
NCT03829436	Nivolumab	TPST-1120			Nivolumab + TPST-1120	I	DLTs MTD	Recruiting
NCT03241173	Nivolumab		INCAGN2385 Ipilimumab		Nivolumab + INCAGN2385 INCAGN2385 + Ipilimumab Nivolumab + INCAGN2385 + Ipilimumab	I/II	Safety and Tolerability ORR	Completed
NCT03667716	Nivolumab		COM 701		COM 701 COM 701 + Nivolumab	I	Safety MTD	Recruiting
NCT03435640	Nivolumab		Bempegaldesleukin NKTR-262		NKTR-262 + Bempegaldesleukin NKTR-262 + Bempegaldesleukin + Nivolumab	I/II	Safety ORR	Active not recruiting
NCT02983045	Nivolumab		Ipilimumab Bempegaldesleukin		Nivolumab + Bempegaldesleukin Nivolumab + Ipilimumab Bempegaldesleukin	I/II	ORR	Active not recruiting
NCT02499367	Nivolumab	Cisplatin Doxorubicin Cyclophosphamid Cisplatin	RT Doxorubicin Doxorubicin Cyclophosphamid Cisplatin		Nivolumab + RT Nivolumab + Doxorubicin Nivolumab + Cyclophosphamid Nivolumab + Cispatin Nivolumab	II	PFS	Active not recruiting
UMIN000030242	Nivolumab	Bevacizumab Paclitaxel			Nivolumab + Paclitaxel + Bevacizumab	II	ORR	Active not recruiting
NCT03952325	Nivolumab Pembrolizumab Atezolizumab	Tesetaxel			Tesetaxel + Nivoluamb Tesexatel + Pembrolizuamb Tesetaxel + Atezolizumab Tesetaxel	II	ORR PFS	Completed
NCT03971409	Avelumab	Binimetinib Utomilumab Liposomal Doxorubicin Sacituzumab Govitecan			Binimetinib + Avelumab Anti-OX40 antibody PF-04518600 + Avelumab Utomilumab + Avelumab Avelumab + Binimetinib + Liposomal doxorubicin Avelumab + sacituzumab govitecan Avelumab + Liposomal doxorubicin	III	BORR	Recruiting
NCT04360941	Avelumab	Palbociclib			Avelumab + Palbociclib	II	MTD ORR	Recruiting
NCT02802098	Durvalumab	Bevacizumab			Bevacizumab + Durvalumab	I	Dynamic of peripheral blood mononuclear cells subpopulations PFS OS	Completed
NCT03616886	Durvalumab	Paclitaxel Carboplatin	Oleclumab	Paclitaxel+ Carboplatin+ Durvalumab	Paclitaxel + Carboplatin + Durvalumab + Oleclumab	I/II	AES CBR	Recruiting
NCT03801369	Durvalumab	Olaparib			Durvalumab + Olaparib	II	ORR	Recruiting
NCT03982173	Durvalumab		Tremelimumab		Durvalumab + Tremelimumab	II	ORR	Active, not recruiting
NCT04837209	Dostarlimab	Niraparib			Niraparib + Dostarlimab + Radiation therapy	II	ORR	Recruiting
NCT03742349	Spartalizumab	Capmatinib	Canakinumab Lacnotuzumab NIR178 LAG525		Spartalizumab + LAG525 + NIR178 Spartalizumab + LAG525 + Capmatinib Spartalizumab + LAG525 + MCS110 Spartalizumab + LAG525 + Canakinumab	I	Safety DLTs	Recruiting
NCT03499899	Spartalizuamb	Carboplatin	LAG525		LAG525 + Spartalizumab LAG525 + Spartalizumab + Carboplatin LAG525 + Carboplatin	II	ORR	Completed
NCT04673448	Dostarlimab	Niraparib			Niraparib, Dostarlimab	I	ORR	Recruiting
NCT03579472	Bintrafusp Alfa	Eribulin			Bintrafusp alfa, Eribulin mesylate	I	RP2D Safety	Recruiting
NCT04609215		Carboplatin Gemcitabine	ALECSAT		ALECSAT + Carboplatin + Gemcitabine	I	Safety	Recruiting
NCT01516307		Phosphate Buffer Saline Cyclophosphamide	Vaccine OPT-822/OPT-821	Phosphate Buffer Saline + Cyclophosphamide	OPT-822/OPT-821 + Cyclophosphamide	II	PFS	Completed
NCT02614833		Paclitaxel	Eftilagimod alpha	Paclitaxel+ Placebo	Paclitaxel+ Eftilagimod alpha	I/II	Dose finding PFS	Completed
NCT00179309		Docetaxel	Panvac Vaccine	Docetaxel	PANVAC + Docetaxel	II	PFS	Completed
NCT03066947		Cyclophosphamide	SV-BR-1-GM Interferon-alpha-2b		Cyclophosphamide + SV-BR-1-GM + Interferon-alpha-2b	I/II	Safety	Completed
NCT04129996	Camrelizumab	Nab-paclitaxel Famitinib			Camrelizumab + Nab-paclitaxel+ Famitinib	II	ORR	Active not recruiting
NCT04303741	Camrelizumab	Apatinib Eribulin			Camrelizumab + Apatinib+ Eribulin	II	ORR	Active not recruiting
NCT03394287	Camrelizumab	Apatinib			SHR-1210+ Apatinib daily dosing SHR-1210 + Apatinib intermittent dosing	II	ORR	Completed
NCT04405505	TQB2450	Nab-paclitaxel Anlotinib		Nab-paclitaxel	TQB2450 + Anlotinib	III	PFS	Not yet Recruiting
NCT02936102	FAZ 053 PDR001				FAZ053 single agent FAZ053 + PDR001	I	Safety and Tolerability	Active, not recruiting
NCT03872791	KN046	Nab-paclitaxel			KN046 KN046 + Nab-paclitaxel	I/II	ORR DOR	Active, not recruiting
NCT04085276	Toripalimab	Nab-Paclitaxel		Nab-paclitaxel+ Placebo	Toripalimab + Nab-Paclitaxel	III	PFS	Recruiting
NCT03893955	Budigalimab (ABBV 181)	CBDCA Nab-Paclitaxel	ABBV 927 ABBV 368		ABBV-927 + Nab-paclitaxel + ABBV-368 ABBV-927 + Carboplatin ABBV-927 + Carboplatin+ Budigalimab ABBV-927 + Carboplatin + ABBV-368	I	ORR RP2D	Recruiting
NCT03517488			XmAb20717		XmAb20717	I	Safety and Tolerability	Recruiting
NCT03752398			Ipilimumab XmAb20717		XmAb20717 XmAb20717 + Ipilimumab	I	Safety and Tolerability	Recruiting
NCT03849469	Pembrolizumab		XmAb22841		XmAb22841 XmAb22841 + Pembrolizumab	I	Safety and Tolerability	Recruiting
NCT03538028			INCAGN2385		INCAGN02385	I	Safety and Tolerability	Completed
NCT04254107	Sasanlimab		SEA-TGT		SEA-TGT SEA-TGT+ Sasanlimab	I	Safety and Tolerability	Recruiting
NCT03665285			NC318		NC318	I	Safety and Tolerability MTD	Recruiting

CT, chemotherapy; ORR, objective response rate; CRR, complete response rate; CR, complete response; PR, partial response; CBR, clinical benefit rate; DCR, disease control rate; DLTs, dose-limiting toxicity; ITT, intention-to treat; PFS, progression-free survival; OS, overall survival; MTD, maximum tolerated dose; RP2D, recommended phase 2 dose; AEs, adverse events; BORR, best overall response rate; CPS, combined positive score; PDL-1, programmed death-ligand 1.

**Table 2 cancers-14-02102-t002:** List of completed or ongoing clinical trials with immune checkpoint inhibitors alone or with combinational drugs and novel immune-modulating strategies for early TNBC treatment.

NCT	Anti PD-1/PD-L1	Non-IT Drugs	IT Drugs	Compartor Arms	Experimental Arms	Phase	Primary Endpoints	Status
NCT04427293	Pembrolizumab	Lenvatinib			Lenvatinib + Pembrolizumab	I	Effectiveness	Recruiting
NCT03639948	Pembrolizumab	Carboplatin Docetaxel			Pembrolizumab + CT	II	pCR rate	Recruiting
NCT04373031	Pembrolizumab	Epirubicin Cyclophosphamide Taxanes	IRX-2	Pembrolizumab+ CT	Pembrolizumab + IRX-2 + CT	II	pCR rate	Recruiting
NCT05203445	Pembrolizumab	Olaparib			Olaparib + Pembrolizumab	II	pathologically negative MRI-guided biopsy after 12 weeks of treatment	Recruiting
NCT02954874	Pembrolizumab			No treatment	Pembrolizumab	III	iDFS Severity of Fatigue Physical function	Active not recruiting
NCT02957968	Pembrolizumab	Decitabine Doxorubicin Cyclophosphamide Paclitaxel Carboplatin			Pembrolizumab + Decitabine Doxorubicin Cyclophosphamide Paclitaxel Carboplatin	II	Dynamic of TILs	Recruiting
NCT05177796	Pembrolizumab	Panitumumab Paclitaxel Carboplatin Doxorubicin Cyclophosphamide			Panitumumab + Pembrolizumab + Paclitaxel + Carboplatin + Doxorubicin + Cyclophosphamide	II	pCR Rate	Active not yet recruiting
NCT03036488	Pembrolizumab	Carboplatin Paclitaxel Doxorubicin/Epirubicin Cyclophosphamide		Placebo+ Carboplatin Paclitaxel Doxorubicin Epirubicin Cyclophosphamide	Pembrolizumab + CBDCA Paclitaxel Doxorubicin/Epirubicin Cyclophosphamide	III	pCR Rate EFS	Active, not recruiting
NCT01986426	Pembrolizumab		LTX-315		LTX-315 + Pembrolizumab	I	DLT	Completed
NCT03197935	Atezolizumab	Nab-paclitaxel Doxorubicin Cyclophosphamide		Placebo+ Nab-paclitaxel+ Doxorubicin+ Cyclophosphamide	Atezolizumab + Nab-paclitaxel Doxorubicin + Cyclophosphamide	III	pCR in ITT pCR in PD-L1+	Completed
NCT03498716	Atezolizumab	Paclitaxel Dose-dense Doxorubicin/Epirubicin Cyclophosphamide		Placebo+ Paclitaxel+ Dose-dense Doxorubicin or Epirubicin+ Cyclophosphamide	Atezolizumab + Paclitaxel Dose-dense Doxorubicin/Epirubicin Cyclophosphamide	III	iDFS	Recruiting
NCT03371017	Atezolizumab	Gemcitabine Capecitabine Carboplatin		Placebo+ Gemcitabine+ Capecitabine+ Carboplatin	Atezolizumab + Gemcitabine + Capecitabine + Carboplatin	III	OS	Recruiting
NCT03256344	Atezolizumab		Talimogene laherparepvec		Talimogene + Laherparepvec + Atezolizumab	I	DLTs	Completed
NCT04102618	Atezolizumab		Pelareorep		Atezolizumab + Pelareorep	I	CelTIL Score	Recruiting
NCT03356860	Durvalumab	Paclitaxel Epirubicin Cyclophosphamide		Paclitaxel+ Epirubicin+ Cyclophosphamide	Durvalumab + Paclitaxel Epirubicin + Cyclophosphamide	I/II	pCR Rate Safety	Recruiting
NCT05209529	Durvalumab	Olaparib			Durvalumab + Olaparib Olaparib monotherapy	II	pCR Rate	Recruiting
NCT02489448	Durvalumab	Nab-paclitaxel Doxorubicin Cyclophosphamide			Durvalumab + Nab-paclitaxel Doxorubicin + Cyclophosphamide	I/II	pCR Rate	Active not recruiting
NCT03740893	Durvalumab	AZD6738 Olaparib		Standard CT	AZD6738 monotherapy Olaparib monotherapy Durvalumab monotherapy	II	Safety Immunomodulating action	Recruiting
NCT03594396	Durvalumab	Olaparib			Durvalumab + Olaparib	I/II	Changes in tumor biology	Active not recruiting
NCT02685059	Durvalumab	Nab-Paclitaxel Epirubicin Cyclophosphamide		Placebo+ Nab-Paclitaxel Epirubicin Cyclophosphamide	Durvalumab + Nab-Paclitaxel Epirubicin + Cyclophosphamide	II	pCR Rate	Completed
NCT01042379	Pembrolizumab Cemiplimab Dostarlimab Durvalumab	AMG 386 Trastuzumab AMG 479 (Ganitumab) Metformin MK-2206 AMG 386 T-DM1 Pertuzumab and Trastuzumab Ganetespib ABT-888 Neratinib PLX3397 Talazoparib Irinotecan Patritumab SGN-LIV1A Olaparib SD-101 Tucatinib REGN3767 Trilaciclib Paclitaxel Encequidar Carboplatin Oral Paclitaxel		Standard Treatments depending on HR/HER2-status	Experimental agents added to standard neoadjuvant treatment	II	pCR rate	Recruiting
NCT04613674	Camrelizumab	Standard CT		Placebo+ Standard CT	Camrelizumab + Standard CT	III	pCR Rate	Recruiting
NCT04301739	HLX 10	Nab-Paclitaxel Carboplatin Doxorubicin Cyclophosphamide		Placebo+ Nab-Paclitaxel Carboplatin Doxorubicin Cyclophosphamide	HLX 10 + Nab-Paclitaxel + Carboplatin Doxorubicin + Cyclophosphamide	III	pCR Rate	Not yet recruiting
NCT03815890	Nivolumab		Ipilimumab		Nivolumab Nivolumab + Ipilimumab	II	Immune activation after pre-operative Nivolumab	Recruiting
NCT03487666	Nivolumab	Capecitabine		Capecitabine	Nivolumab Nivolumab + Capecitabine	II	Immune activation measured by changes in the peripheral immunoscore (PIS) at week 6	Active not recruiting
NCT03818685	Nivolumab	Capecitabine	Ipilimumab	Capecitabine	Nivolumab + Ipilimumab	II	iDFS	Recruiting
NCT04185311	Nivolumab		Ipilimumab+ Talimogene laherparepvec		Talimogene laherparepvec + Nivolumab + Ipilimumab	I	Safety	Active not recruiting
NCT02938442		Doxorubicin Cyclophosphamide Paclitaxel	P10s-PADRE with MONTANIDE™ ISA 51 VG	Doxorubicin + Cyclophosphamide Paclitaxel	P10s-PADRE with MONTANIDE™ ISA 51 VG + Doxorubicin + Cyclophosphamide + Paclitaxel	I/II	Safety pCR Rate	Recruiting
NCT02779855		Paclitaxel	Talimogene laherparepvec		Talimogene laherparepvec + Paclitaxel	I/II	MTD RP2D pCR Rate	Active not recruiting

CT, chemotherapy; pCR, pathological complete response; MRI, magnetic resonance imaging; iDFS, invasive disease-free survival; TILs, tumor-infiltrating lymphocytes; EFS, event free survival; DLTs, dose-limiting toxicity; ITT, intention-to treat; OS, overall survival; MTD, maximum tolerated dose; RP2D, recommended phase 2 dose.

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
