# Peer review of "Immune-Based Therapy in Triple-Negative Breast Cancer: From Molecular Biology to Clinical Practice"

_cancers, 2022, doi:10.3390/cancers14092102_

Round 1

Reviewer 1 Report

Authors aims to summarize the recent knowledge on BC immune-gram and the 38 biomarkers proposed to support ICIs based therapy in TNBC, and to provide an overview of po-39 tential strategies to enhance immune response and overcome mechanisms of resistance. I enjoyed reading this review and thinks that this will be very helpful for the growing field. I do not have any major objections with this review. Nice work!. Thank you!

Author Response

Dear reviewers,

Thank you for taking the time to review our manuscript.

We are pleased with the positive comments.

The paper has been carefully reviewed for improvement of the English language.

We are sending you the revised manuscript with all the highlighted changes for your review.

We hope you all find them with your satisfaction.

Thanking you in advance, I look forward to hearing from you.

Best regards,

Francesca Carlino, on behalf of all authors

Reviewer 2 Report

This is a comprehensive review in the area of immunotherapy for triple negative breast cancer. The subject is timely and important for the readership.

The main question is the mechanism and effectiveness of immunotherapy in triple negative breast cancer. The topic is relevant in the field, but not original as it is a review article and does present already published data. 

Methodology and references are appropriate.

The conclusions are consistent with the evidence and arguments presented and they address the main question posed.

Author Response

(The authors gave the same response as above.)

Reviewer 3 Report

cancers-1623066-peer-review-v1

General Comments: This is one of the most comprehensive and well-written review papers I have read in some time. Therefore, this was a joy to review. No edits/changes requested.

Author Response

(The authors gave the same response as above.)
